# Ground Stress Analysis and Automation of Workface in Continuous Mining Continuous Backfill Operation

**Seun A. Ajayi** [1], **Liqiang Ma** [1,2,*] and **Anthony J. S. Spearing** [1]

1 Key Laboratory of Deep Coal Resource Mining, Ministry of Education of China, School of Mines, China University of Mining and Technology, Xuzhou 221116, China; seunajayi93@gmail.com (S.A.A.); ajsspearing@yahoo.com (A.J.S.S.)
2 School of Energy Engineering, Xi'an University of Science and Technology, Xi'an 710054, China
* Correspondence: ckma@cumt.edu.cn

**Abstract:** The cost, complexity, lack of filling space and time create challenges in the longwall backfill operation, resulting in poor subsidence control and reduced productivity. This paper proposes an automated continuous mining and continuous backfill (CMCB) method by examining its key requirements and investigates the optimum sequence of coal panel (such as drifts) excavation to ensure ground strata control at relatively high productivity. The automated CMCB adopts the highwall mining technique underground, which enables easier automation at the workface. A numerical simulation of the Changxing coal mine in China was undertaken, and five different sequences of coal excavation were investigated, using the automated CMCB excavation parameters (assuming a 4 m width cut, 5 m mining height for a 200 m long coal slice) to determine the optimum sequence of resource excavation. The plastic zones and vertical displacement across the five models were analyzed. Simulation results of the 5 m high coal seam excavation show that the odd-even slice (OES) mining sequence, which has a vertical ground displacement of 74 mm, is the most efficient excavation method, due to its effective stress redistribution and lower induced ground displacement.

**Keywords:** continuous mining and continuous backfilling; highwall mining; hydraulic support; rock strata stress; underground mining



## 1. Introduction

China is a mineral resources-rich nation, with enormous reserves of lead-zinc estimated at 849 million tons, manganese at 4.8 billion tons, lithium chloride at 92.48 million tons [1]. It has the world's third-largest reserve of coal and is the largest producer of gypsum [1]. These mineral deposits span numerous provinces, and underground mining accounts for a substantial percentage of tonnage produced. Due to the overburden depth of the reserves, underground mining accounts for 95% of tonnage output [2]. The tonnage of coal resources with less than 600 m overburden is about 20% of the total reserves in China [3]. As a result, the vast majority of China's coal reserves cannot be exploited by the surface mining operation, and the longwall mining method accounts for 90% of its coal production [4].

The deep mining depth of ore reserves makes room and pillar impractical, due to the very low percentage of possible extraction. The main challenges of the underground mining methods are coal bursts, mine accidents, land subsidence, water aquifer pollution, air pollution, groundwater table displacement, acid mine drainage, disturbance of hydro-geology, displacement and relocation of villages and farmlands [2,5]. These adverse impacts and their mitigations have been extensively studied in various countries, including Poland [6–9], Australia [10,11], UK [12], South Africa [13,14] and India [15,16]. In China, the environmental impact of land subsidence due to underground mining operations is severe; about one million hectares of subsided land exist today [2]. In areas with mineral reserves under buildings, railways, and water bodies, which are also referred to as under

three bodies [17], land subsidence poses environmental challenges, socio-economic and mining layout challenges [18].

Reducing these adverse impacts of underground mining has necessitated the adoption of various mining techniques; the most significant has been the incorporation of backfilling into mining operations. The application of backfill materials and backfill technologies has successfully mitigated land subsidence with backfill methods, such as solid backfill mining, hydraulic backfilling, and paste backfilling methods with subsidence factors ranging from 0.1 to 0.3 [18]. However, due to limited supported and protected space and lack of filling time, the backfilling operations in longwall mining affect resource productivity. They do not effectively control the movement of overlying strata, especially on a surface with critical infrastructures.

As a result, the continuous mining continuous backfill method was proposed. Notable mining methods that have incorporated the continuous mining and backfill technique include the Wongawilli roadway backfilling coal mining method (WRBCM) [19], longwall mining with split-level gate roads (LMSG) [20], and continuous mining continuous backfill method (CMCB) [21]. These methods use the cemented paste backfill technique with readily available backfill materials, such as aeolian sand and fly ash. These methods have been successful in the mitigation of subsidence. However, the conventional continuous mining continuous backfill operation requires human presence in the workface for excavation operations. This is a drawback to conventional CMCB operation, and it further exacerbates the existing automation challenges in longwall mining operations.

To solve the existing longwall mining challenges, China is implementing intelligent mining technology in coal mines nationwide [22,23] to achieve much safer, highly automated and efficient mining [24,25]. To achieve intelligent mining, it is necessary to deploy highly automated mining equipment at the working face [26]. This paper aims to identify a mining method that enables easier automation at the workface with minimal ground displacement and allows for a higher flexibility in the excavation of resources at a lower capital cost. To achieve this aim, the highwall mining technique of remote deployment of equipment at the mine workface is combined with the effective subsidence control merit of a conventional continuous mining and continuous backfill. This concept is herein termed automated continuous mining continuous backfill.

The excavation parameters of the automated CMCB are used in numerical simulation to determine the most effective sequence of coal excavation in a typical mine. While our research team has proposed this mining method in an earlier publication [27], this paper further enumerates the equipment requirements, modifications and positioning, ground stress and mine layout. The mining equipment and mine layout were drawn using Creo Parametric and AutoCAD. Geological data of a typical underground mine in the western part of China published by Zhang, et al. [28] were used for the FLAC$^{3D}$ numerical simulation. The induced ground stresses and vertical displacements of the mined area were investigated in five different sequences of mineral excavations.

## 2. Automated Continuous Mining Continuous Backfill Method

The automated continuous mining continuous backfill (CMCB) method adopts the highwall mining principle for the underground exploitation of resources. The addcar highwall mining system is modified for underground mining operations, due to its advantage over the auger and bridge conveyor system Walker [29]. The conventional addcar system employs continuous miners and addcars that utilize screws to remove coal from the entry.

The major equipment required for a successful automated CMCB mining operation includes continuous miners, a hydraulic ram support system, addcars, an armored face conveyor system, a monorail and decouple machine. The decouple machine is a newly introduced equipment for addcar coupling and decoupling. These machines will be mechanically linked together and mounted with sensors to achieve a high level of autonomous mining. The proposed mining technique aims for a fully automated mining methodology or automation at a level higher than the longwall method.

### 2.1. Automated CMCB Operation Technique

As shown in Figure 1, the continuous miner with a 4 m width cutting drum makes the first cut into the coal slice along the x-axis; as the continuous miner advances in the y-direction, successive addcars are attached to it with the aid of the coupling-decoupling machine. The telescopic cylinder of the hydraulic support system pushes the addcar; thus, the addcars act as a drill rod for the continuous miner and as a conduit to convey coal back out to the armored face conveyor. This operation continues until the complete coal slice (200 m length) is mined out, and then the mining equipment is removed and moved to the next coal slice. A monorail is installed on the mine roof, and it connects the addcars stack (i.e., storage for 50–60 addcars) to the coal workface. In addition, the couple-decouple machine is mounted on the monorail in front of the hydraulic support system.

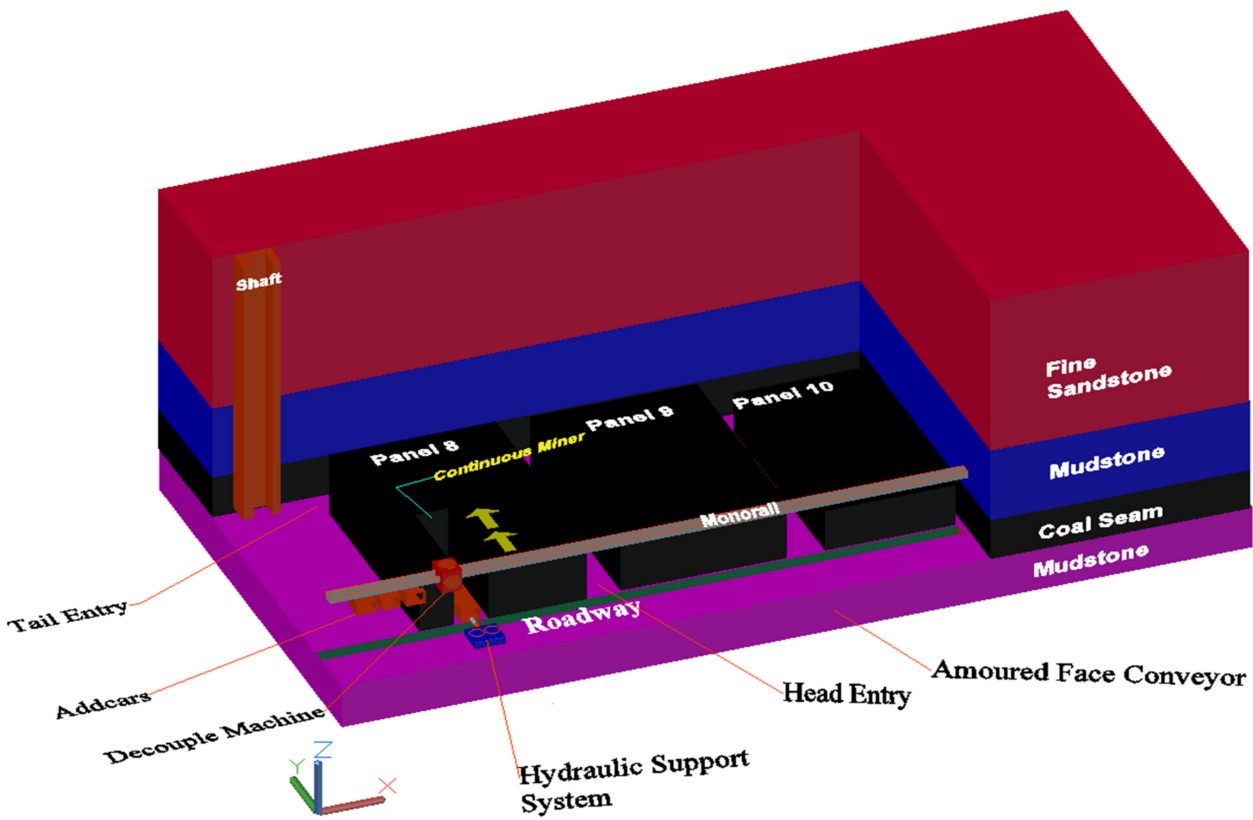

**Figure 1.** Automated CMCB mining method (performed by the authors).

There is a considerable knowledge base from conventional surface highwall operations and some of the calibrated modelling for deeper underground conditions [19,21]. According to Ma et al. [19], the key to the success of a remote operation is mining at a stable, short-term unsupported span. In addition, according to Spearing et al. [27], highwall underground mining should be conducted up-dip to facilitate water drainage and make backfilling easier and less costly. The backfilling should be carried out downdip to improve roof contact. The equipment suite for underground needs to be much more compact than for surface operations. As mining progresses towards the main entries, the monorail sections can be removed and reinstalled in front of the retreating overall panel face (towards the main entries) as successive slices are mined [27].

### 2.2. Mine Design Layout and Equipment Positioning

Mine design will be carried out based on the geological and geotechnical conditions of the individual mine; therefore, the type of entry is not predetermined. Relative positioning of equipment is shown below in Figure 2.

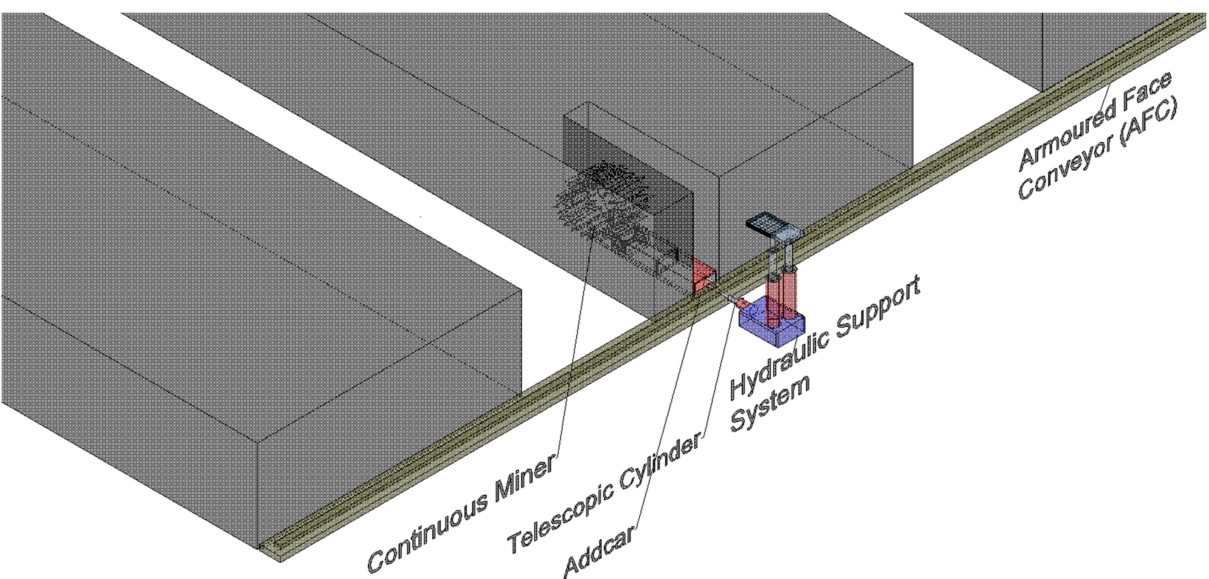

**Figure 2.** Mine equipment positioning (performed by the authors).

## 2.3. Estimated Production for CMCB

The estimated production rate per coal slice of 200 m long, 4 m wide, with 5 m in height has a coal-specific gravity of 1.3 at a 165 tons per hour production rate for one continuous miner. For the available time of 400 min (approximately 6.7 hrs available in 8 hrs per shift), a continuous miner will produce 1105 tons per 8 hr-shift. On a conservative estimation, 3 shifts of 8 h of production per shift result in 1105 × 3 = 3315 tons per day. Using 828 shifts per year, at 6 days per week and 46 weeks per year, a single continuous miner will produce approximately 915,000 tons per year. The application of a super-section, i.e., the use of multiple continuous miners, will have a multiplying effect on the output and increase the cutting capacity of the system, which will significantly increase the tonnage per year.

## 3. Automated CMCB Equipment Requirements

### 3.1. Continuous Miner

A continuous miner is made of a large rotating steel drum with tungsten carbide teeth. This means that continuous miners are suitable for cutting soft and some harder rock deposits, such as potash, coal, gypsum, manganese, limestone, salt, and trona. Several designs of continuous miners have been manufactured, such as the Joy Ripper miner, boring miner, milling-head miner, auger miner, and the boom-type miner. The CMCB mining method does not require shuttle cars, so the continuous miner chain conveyor needs to be modified and connected to an addcar, as is the case with a conventional highwall addcar system, to ensure a continuous haulage operation.

### 3.2. Armoured Face Conveyor (AFC)

AFC is a one-sided trough scraper conveyor, invented by the Caterpillar equipment manufacturing company in the early 1940s, with the second side of the trough being formed by the coal face [30]. Modification of the AFC will include side holes that act as an opening for the hydraulic ram support's telescopic cylinder push and pull function. The overall capacity reduction in a typical armored face conveyor is required because the tons per hour between 1433 to 5512 tons of a typical AFC is excessive for the CMCB, except when multi-section mining or multiple continuous miners are simultaneously used during the excavation operation. In addition, the AFC acts as a guide rail for shearer with a traction force of up to 1000 kN in a longwall mining operation; this feature is not required for the

CMCB operation. Hence, capacity reduction and removal of guide rail functionality of typical AFC increases the compatibility of the equipment required and reduces capital cost.

### 3.3. Hydraulic Support System

Since the first hydraulic support was developed in England in 1854, hydraulic supports have been essential equipment in mechanized coal mining systems [31]. Presently, the highest working height of hydraulic support globally is 8.8 m [32]; however, the required design for the CMCB operation is a medium seam, slicing mining hydraulic support with a maximum design height of 5 m.

A modification requirement for the hydraulic support is the integration of a telescopic cylinder shown in Figure 3—a hydraulic powered telescopic cylinder connected to the base of the hydraulic support to facilitate the push–pull mechanism of the addcar. In addition, a signal receiver is mounted on the hydraulic support to aid communication across the addcar, decouple machine and hydraulic support. A caving shield in convectional hydraulic support is not required. The parameters of hydraulic support suitable for medium height coal seam is presented below in Table 1.

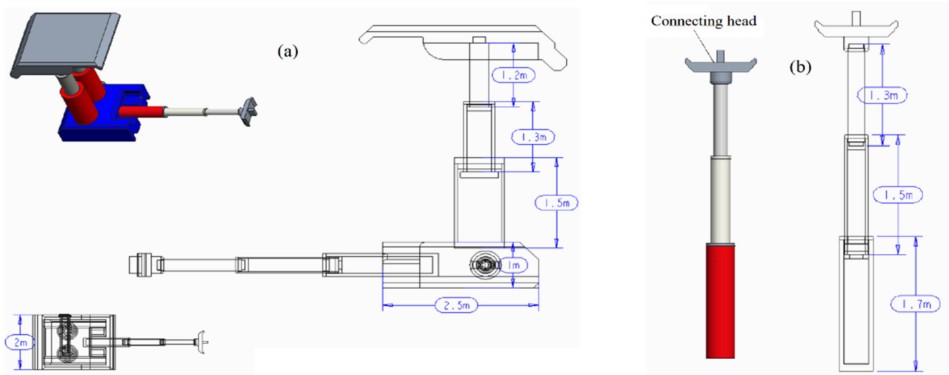

**Figure 3.** Equipment dimension. (**a**) Two-stage hydraulic support; (**b**) telescopic cylinder (performed by the authors).

**Table 1.** The hydraulic support system parameters.

| Mechanical Features | Type/Values |
| --- | --- |
| Support Type | Two-Leg Hydraulic Support |
| Cylinder Type | Two-Stage Hydraulic Cylinder |
| Pump pressure | 31.5 MPa |
| Setting Load (31.5 MPa) | 3969 kN |
| Operating Height | 5 m |
| Collapse/Retract Height | 2.5 m |
| Base Size | 2.5 m × 2 m |
| Assumed Weight | 20 tons |

Setting load = area of bottom (first) stage cylinder × pump pressure.

The functionality requirement of the hydraulic support system for the CMCB method is to provide roof support and, secondly, provide push and pull force for the addcar by using the telescopic cylinder attached to the hydraulic support base. The telescopic cylinder has a connecting head that fits into the back connector of the addcar.

Considering stiffness, which is a factor of operating height (4 m) and cylinder diameter (0.4 m), these collapse heights and operation heights are selected for the design to avoid a reduction in the load-carrying capacity. In addition, the cylinders' diameters are constrained by stiffness and a reduction or increase in the diameters will significantly affect the load-

carrying capacity. The two-stage hydraulic system has been selected due to its relatively higher load-carrying capacity than a three-stage system at the same operating height.

### 3.4. Addcar

The addcar highwall mining system, which was first operated at the Boomer mine in Fayette County, West Virginia in 1990, has since mined over 120 million tons globally [33]. A typical addcar of the addcar system is a rectangular-shaped body, with a screw or belt powered by an electric motor to convey from the continuous miner.

The addcars need to function as a conveyor and a drill rod to push and pull the continuous miner and act as a coal conveyor from the continuous miner to the AFC. The modification required for a typical addcar as shown in Figure 4, is the addition of two connectors to its ends, which will enable coupling to the hydraulic support at one end and the successive addcar at the opposite end.

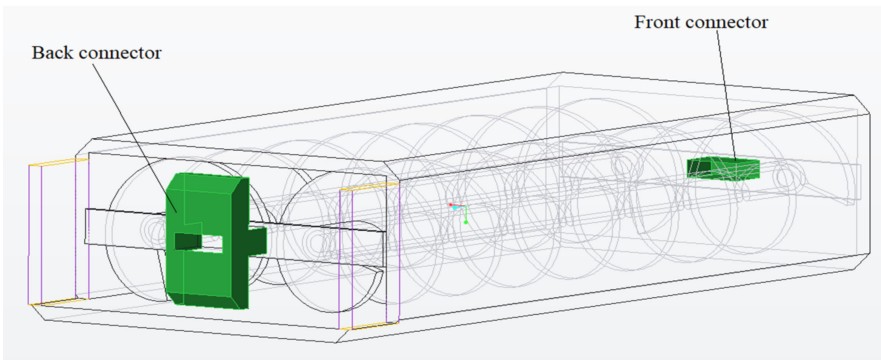

**Figure 4.** Addcar (performed by the authors).

The estimated weight of a single addcar is 8 tons, and the calculated loaded capacity for a single addcar is 10.4 tons. Achieving a 200 m long slice penetration requires 50 addcars. Each addcar has a dimension of 4 m × 2 m × 1 m, length, width and height, respectively. These addcars will be stacked at the main entry in the addcar stack station and moved using the monorail system. The hydraulic support is integrated with the addcar in Figure 5.

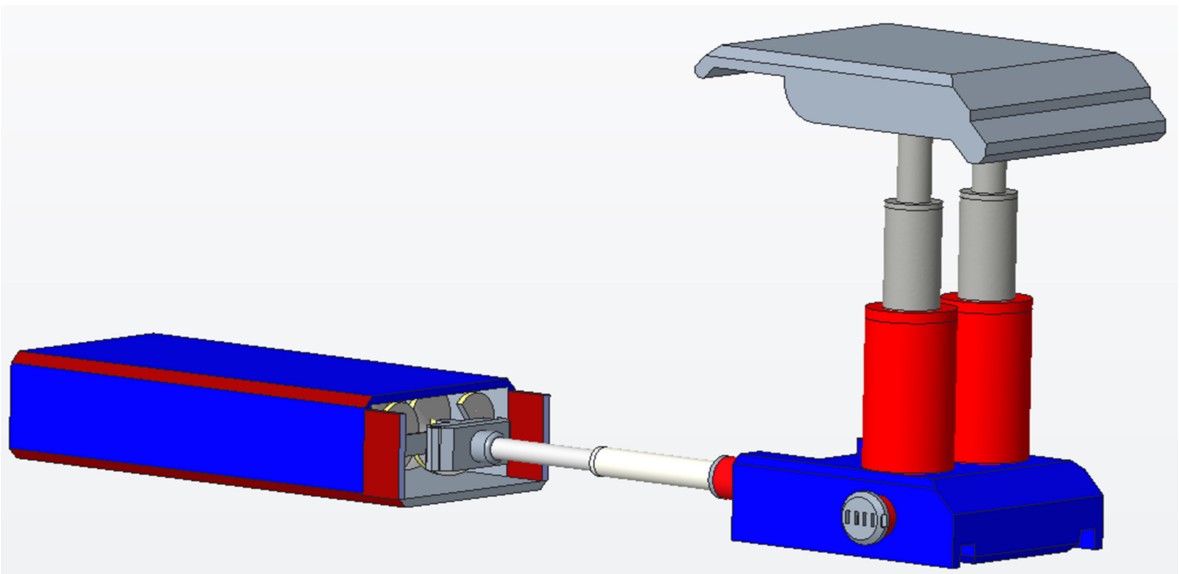

**Figure 5.** Hydraulic support integrated with addcar (performed by the authors).

### 3.5. Monorail System

Monorail technology has been utilized in the mining industry since the late 19th century [34]. Currently, monorails are used in underground coal and ore mining to transport, batch haulage, and manage services to the longwall face.

A typical monorail locomotive is coupled with hydraulic lifting units (with load capacities ranging from 8 t to 50 t), on which containers for material or a cabin for people are suspended. These containers are replaced with addcars for the CMCB operation. The monorail connects from the addcar rack station to the mine panels. The functionality requirement of the monorail is the transportation of addcars from the stack station to the current workface.

## 4. Numerical Simulation of Induced Ground Stress and Displacement

The mine data used for the FLAC$^{3D}$ simulation are typical underground coal mine data extracted from Zhang et al.'s [28] publication, with the minefield's geological, physical, and mechanical properties given below.

### 4.1. Mine Overview

The Changxing coal mine is located in Yuyang district, about 15 km north of Yulin, Shaanxi Province, China. The coal mine covers a field area of 4.82 km$^2$. The primary mineable coal of Changxing coal mine is the number 3 coal seam, which has a stable horizon and simple structure. This coal seam has an average thickness of 5.35 m and belongs to the stable ultra-thick seam. The burial depth of the minefield gradually increases from southeast to northwest, with the coal seam slightly tilting to the northwest at an average angle of 0.5° (Table 2). The overburden depth of the coal seam is 130 m, which is a typical shallow buried coal seam in the western part of China.

**Table 2.** Geological features of the deposit.

| Geological Features | Values |
| --- | --- |
| Average overburden depth | 130 m |
| Coal seam height | 5.35 m |
| Dip angle | 0.5° |
| Mining method | Roadway backfill coal mining (RBCM) |

The soil profile of the coal seam overburden has medium-fine granular arkose (sandstone) as the main roof, with a thickness of 4.48–33.2 m. The immediate roof is silty mudstone; the main floor is siltstone with a thickness of 0.10–9.28 m; and the coal seam has simple structure with strong compressive strength, which hardly causes floor heaving.

This is a typical geological feature of a non-inclined coal minefield in western China, which accounts for more than 85% of the total coal deposit.

### 4.2. Numerical Simulation

The FLAC$^{3D}$ is a three-dimensional finite-difference program developed by the Itasca company. It is one of the most widely used numerical simulation software in geotechnical engineering, such as underground mining [35]. FLAC$^{3D}$ simulation of the coal mine was created as shown in Figure 6 below, with an overburden rock profile of 130 m depth, coal seam of 5 m thickness, floor of 15 m thickness, and reserve width of 248 m. While the required workface width is 48 m, the 248 m width was modelled with 100 m wide coal pillars at both sides of the workface to avoid the effects of artificial boundaries. The dimension of the simulated coal panel is 200 m × 48 m × 5 m in length, width and height, respectively. No in situ stress was considered, and the gravity was set to −10 m/s$^2$. There were 163,680 zones and 174,250 grid points in the numerical model. The simulated coal panel below shows roof 4, roof 3, roof 2 and roof 1, denoting aeolian sand, loess, gelling agent; medium sandstone; fine sandstone and mudstone, respectively. In addition, floor 1,

floor 2 and floor 3 denote siltstone, fine sandstone and mudstone, respectively, with the coal seam between the roof and floor layers. Lastly, the coal seam is divided into 12 slices, denoted by slice1 to slice12. The physical and mechanical properties of individual rock lithology shown in Table 3 were assigned, and the Mohr–Coulomb mechanical model was applied to the simulation.

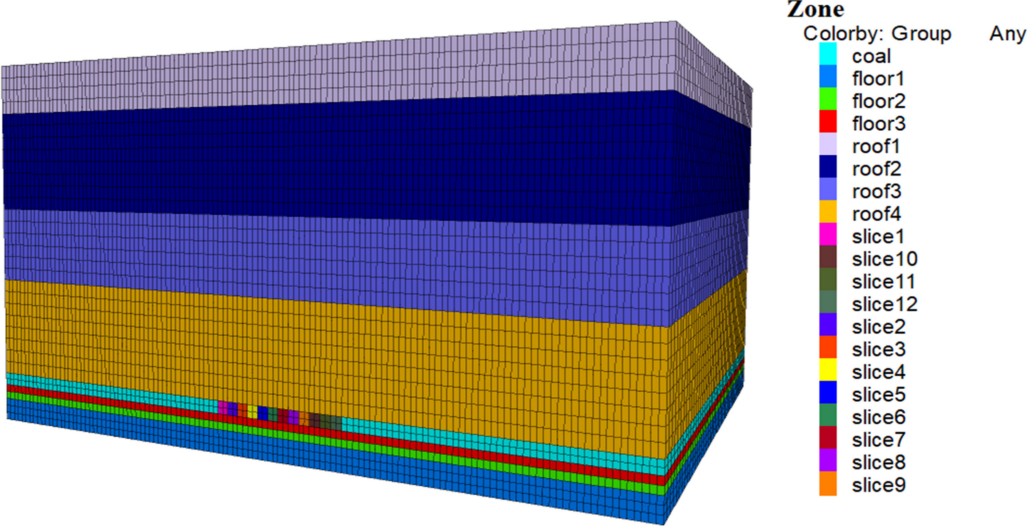

**Figure 6.** Coal panel simulation.

**Table 3.** The physical and mechanical properties of rock lithology in the numerical model.

| Lithology | Bulk Modulus (GPa) | Shear Modulus (GPa) | Cohesion (MPa) | Tensile Strength (MPa) | Internal Friction Angle (°) | Densitykg/m³ |
|---|---|---|---|---|---|---|
| Medium sandstone | 1.8 | 1.1 | 2.2 | 3.2 | 31 | 2100 |
| Fine sandstone | 1.5 | 0.9 | 1.8 | 2.8 | 32 | 2200 |
| Mudstone | 0.8 | 0.5 | 0.9 | 1.6 | 28 | 1600 |
| Coal | 0.6 | 0.4 | 0.8 | 1.2 | 21 | 1400 |
| Mudstone | 1.0 | 0.6 | 0.5 | 1.8 | 28 | 1600 |
| Siltstone | 2.5 | 1.5 | 2.0 | 2.4 | 30 | 2615 |
| Fine sandstone | 1.5 | 0.9 | 1.8 | 2.0 | 30 | 2100 |
| Aeolian sand, loess, gelling agent (1:0.3:0.16) | 0.4 | 0.2 | 0.8 | 0.8 | 27 | 1500 |

*4.3. Simulation Process*

In order to analyze stress distribution and vertical displacement along the coal workface and ground surface, the models' X and Y planes were fixed at both ends, and the Z plane was fixed at the lower plane to allow vertical displacement. The coal slices were excavated in different sequences given in Table 4 below, using the FLAC$^{3D}$ inbuilt mechanical null model, and the mechanical elastic model was applied to backfill excavated stopes. The elastic constitutive model was assigned lower mechanical properties values to mimic a cemented paste backfill mixture. The numerical analysis results are interpreted as first (first six slices) and second (last six slices) excavation cycles, due to significant differences in the simulation results.

Meaning of Models' Nomenclature

The DOS implies a continuous excavation of coal slices 1 to 12 without alternation; this is only practical with the application of web pillars. The OES means the excavation of the odd-numbered slices, while the even-numbered slices remain as pillars and vice-verse

for the EOS (Figure 7). The DGS implies the mining out of a slice, then leaving out the successive two slices to act as pillars and retaining walls for backfill, and lastly, the TGS means excavating a slice, while skipping the successive three slices to act as pillars and retaining walls.

**Table 4.** Slice mining sequence variation.

| S/N | Sequence Name | Slice Mining Order | Initials |
|---|---|---|---|
| 1 | Direct Order Sequence | 1-2-3-4-5-6-7-8-9-10-11-12 | DOS |
| 2 | Odd-Even Sequence | 1-3-5-7-9-11-2-4-6-8-10-12 | OES |
| 3 | Even-Odd Sequence | 2-4-6-8-10-12-1-3-6-7-9-11 | EOS |
| 4 | Double Gap Sequence | 1-4-7-10-2-12-3-6-9-5-8-11 | DGS |
| 5 | Triple Gap Sequence | 1-5-9-2-6-10-3-12-7-11-4-8 | TGS |

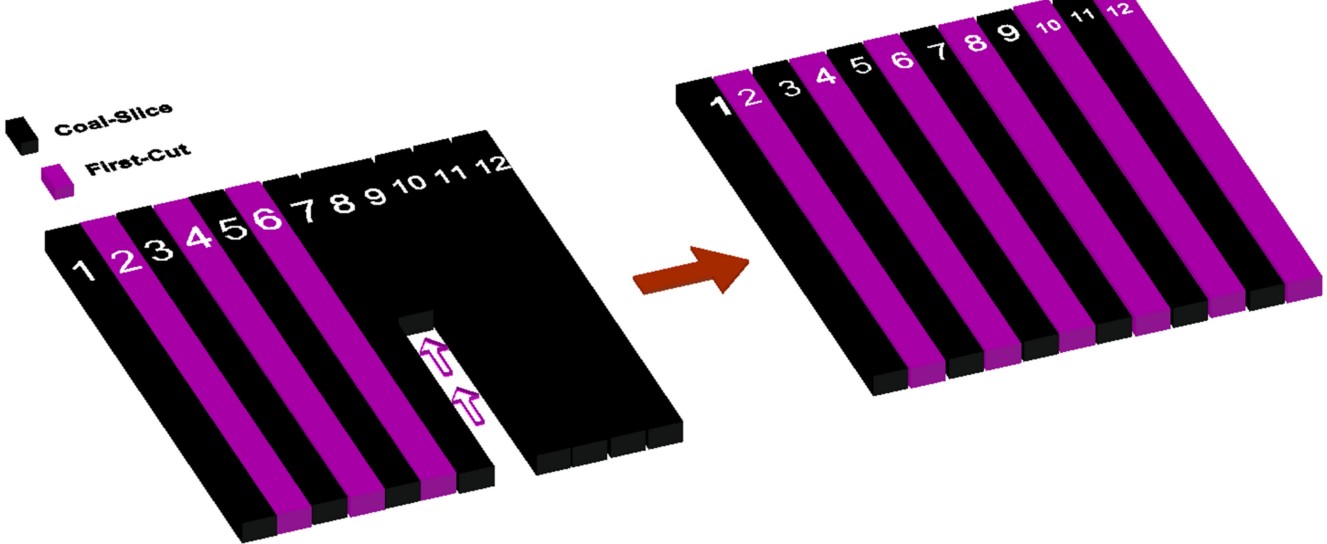

**Figure 7.** Even-odd slice mining sequence.

### 4.4. Contour of Vertical Stress

The DOS simulated result shows the compressive stress of $3.2 \times 10^6$ acting on the surrounding coal pillars and backfilled stopes. The odd-even sequence excavation shows a stress distribution of $3.47 \times 10^6$ along the left out coal slices (2-4-6-8-10-12) acting as temporary pillars, and the even-odd sequence shows a stress distribution ($3.47 \times 10^6$) across the temporary pillars (1-3-6-7-9-11). The double gap sequence stress distribution ($3.5 \times 10^6$) across the temporary pillars (3-6-9-5-8-11) and the triple gap sequence shows a stress distribution of $3.5 \times 10^6$ along the left out coal slices (3-12-7-11-4-8) acting as temporary pillars.

This is shown across the simulations in Figure 8 below at the bottom center, i.e., the 48 m width workface. The direct order sequence mining requires the incorporation of a web pillar to act as a support and retaining wall for the backfill mixture, which will reduce the available time for coal extraction due to the required waiting time for the setting of backfill slurry. The EOS and OES models show that the first six coal slices (2-4-6-8-10-12) and (1-3-5-7-9-11), respectively, can be excavated with appropriate stress redistributions on the surrounding temporary pillars without the requirement of web pillars and waiting time. The DGS model shows that during the first six slice (1-4-7-10-2-12) excavations, coal slice 2 will be excavated next to a backfilled stope, the same as coal slice 5 in the second (3-6-9-5-8-11) excavation cycles.

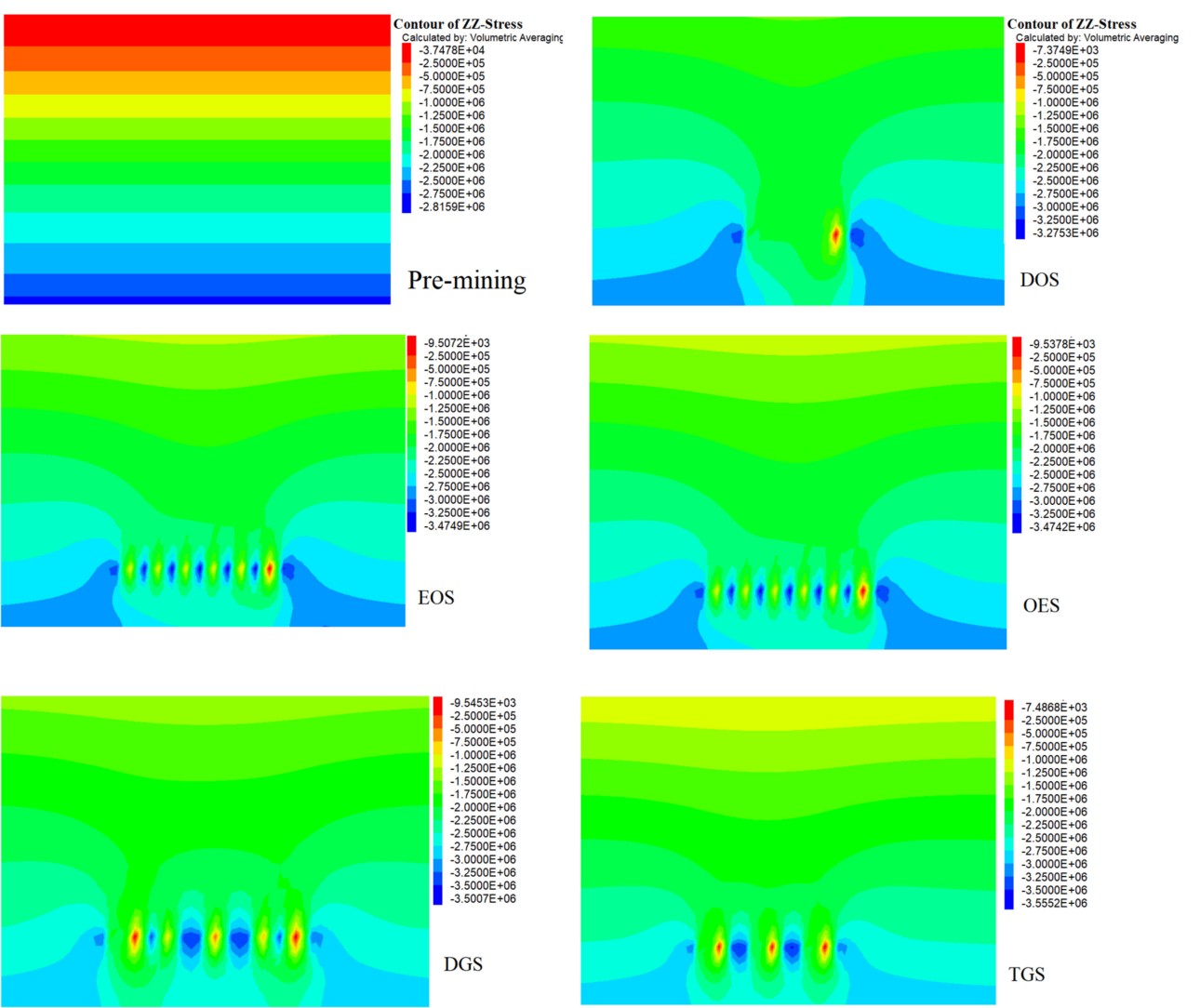

**Figure 8.** Contours of vertical stresses across five models.

Similarly, the TGS model shows that during the first and second excavation cycles, three coal slices (2-6-10) and (11-4-8), respectively, will be excavated adjacent to the backfilled stopes within their respective cycles; therefore, consideration needs to be given to the setting time of the applied backfill slurry.

As shown in Figure 9, the maximum principal stresses in the models occur on the ground surfaces and the immediate roofs. Immediate roofs of the DOS and OES models show the highest and lowest levels of maximum principal stresses, respectively. In addition, the shear strain contour shows that when excavating the coal slices from left to right, the preceding coal pillar or backfill will experience the highest amount of shear deformation.

The zones of maximum shear stresses occur at the backfilled stopes (Figure 10). The EOS and OES undergo the highest shear deformation at $1.4 \times 10^6$.

The zones of shear strain in the overburden show that the D0S ($7.0 \times 10^{-9}$) and the TGS ($5.0 \times 10^{-9}$) experience the highest and lowest shear strain, respectively (Figure 11).

*4.5. Ground Subsidence and Displacement*

In the first excavation cycle, the vertical displacements across the models were examined using the profile of z-displacement for the ground surface at 20 m depth and the excavation roof at 130 m depth. In the Figure 12a,b below, among the five models, the TGS

model shows the least ground surface and immediate roof displacements at 44 mm and 48 mm, respectively.

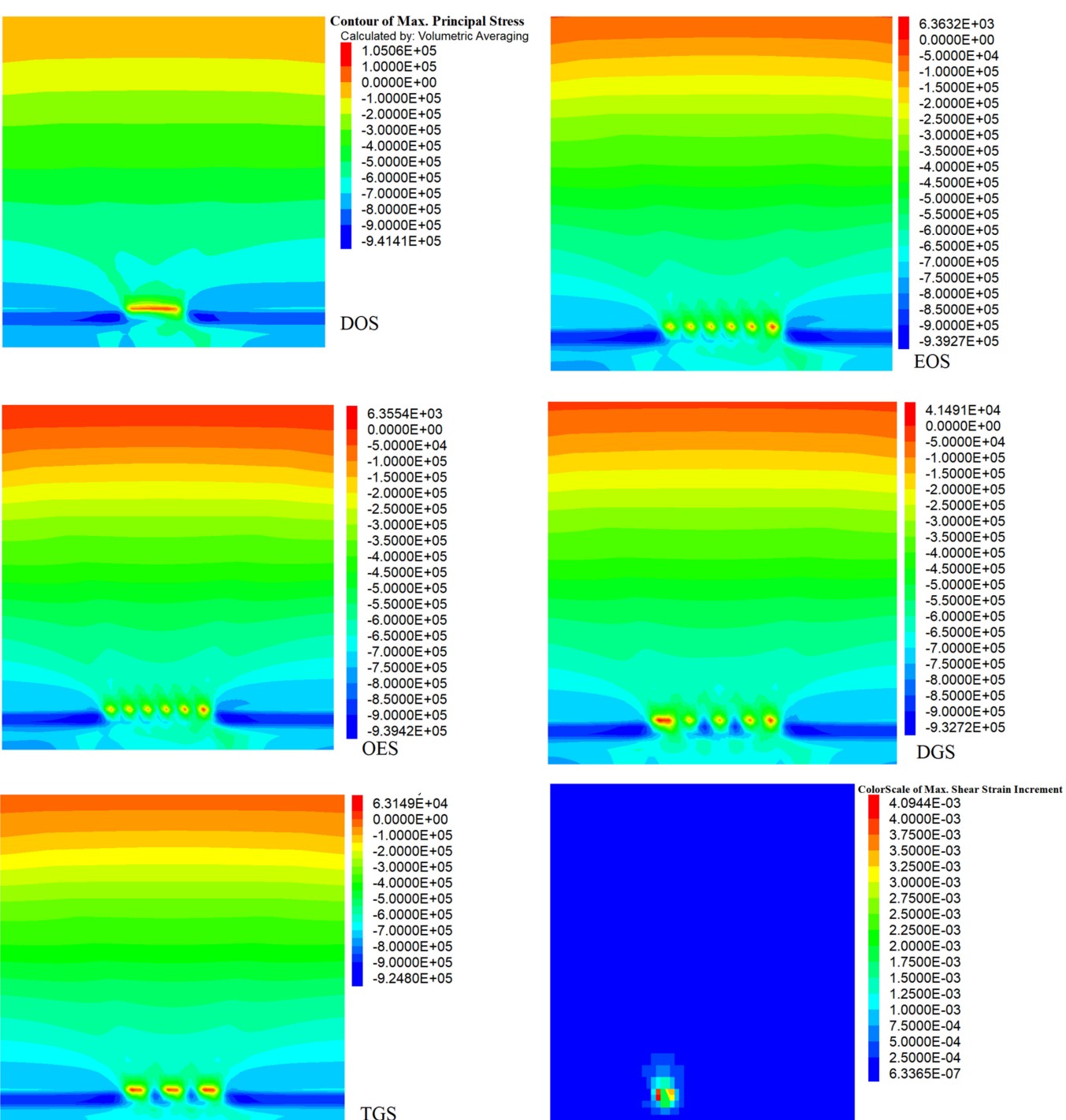

**Figure 9.** Contours of maximum principal stresses.

However, during the second excavation cycle, excavating the last six coal slices in a given panel, the displacements across the models decrease, indicating that the backfill bodies became active in supporting the overlying load. These displacement values converge towards the ninth coal slice excavation across the five models (Figure 12c). Ground displacements across the models are reduced to nearly 25 mm when the panel's last coal slice is excavated and backfilled (Figure 12d). Therefore, induced ground displacement peaked

across the models during the first excavation cycle, with TGS model ground displacement at 44 mm and EOS ground displacement of 75 mm.

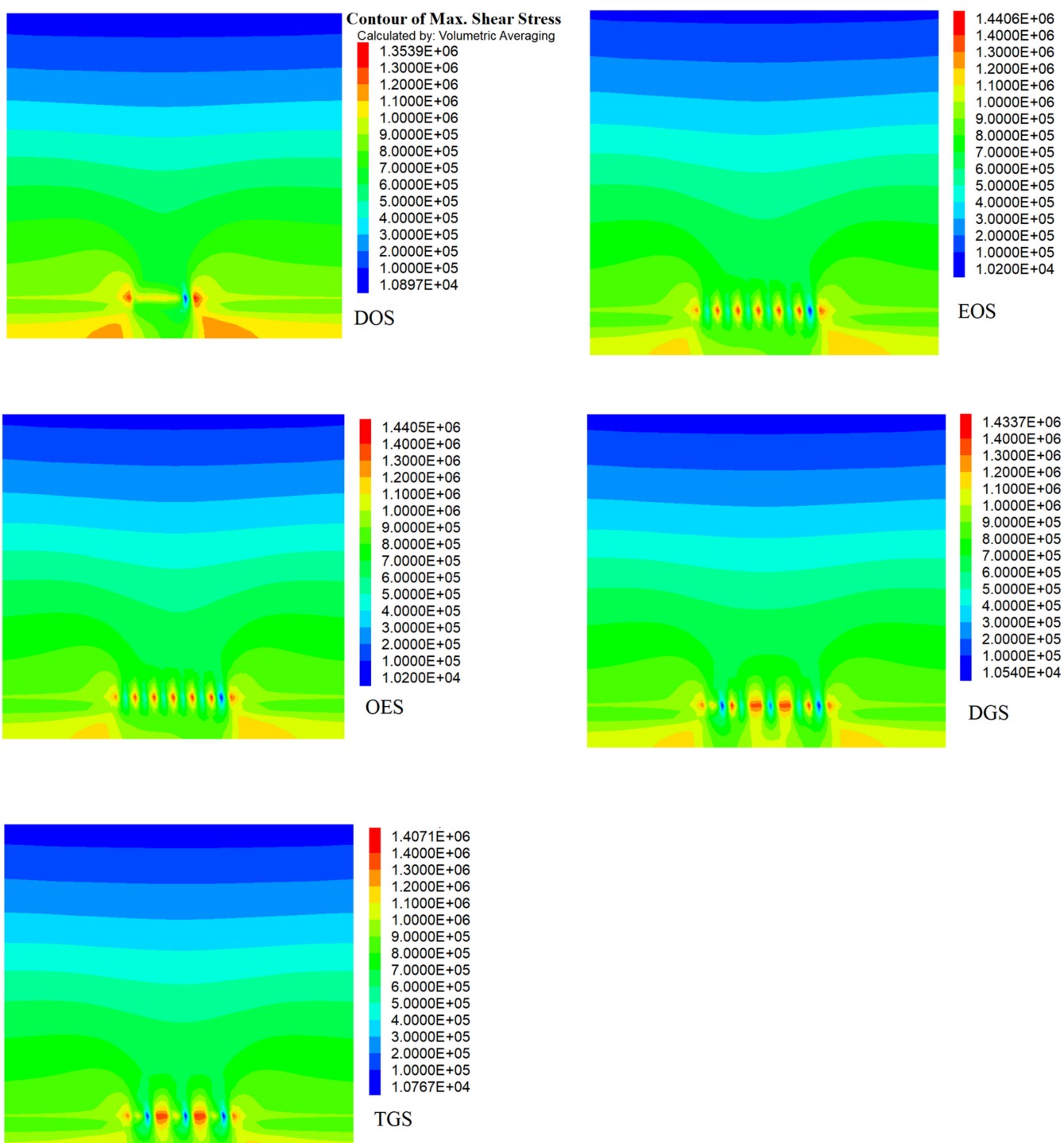

**Figure 10.** Contours of maximum shear stresses.

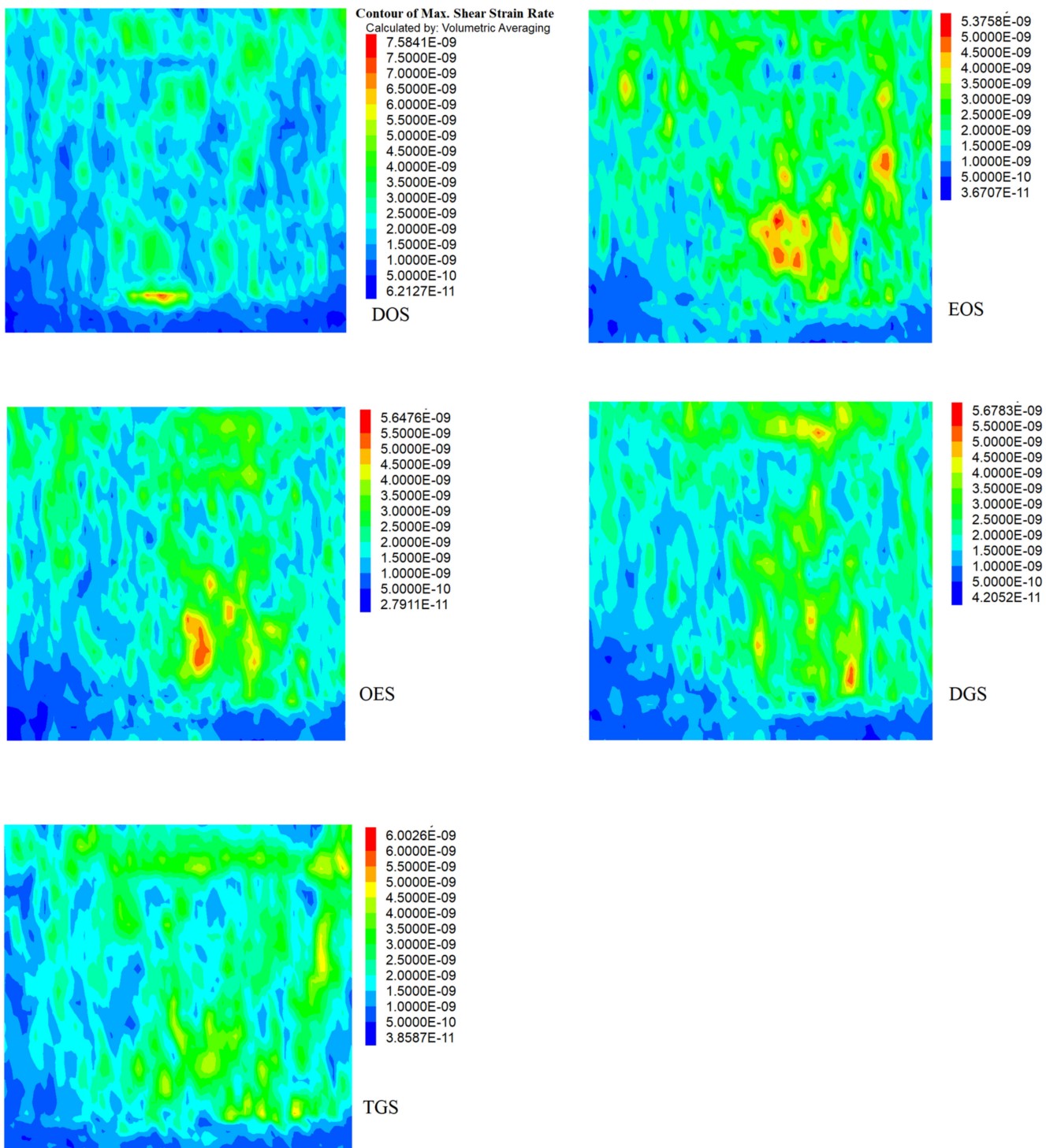

**Figure 11.** Contours of maximum shear strain rates.

The numerical simulation shows that ground displacement for a particular panel excavation will not continuously increase throughout the excavation operation, but peak at some point when the backfill bodies become active in bearing the overlying load and the ground displacement will start to reduce and stabilize.

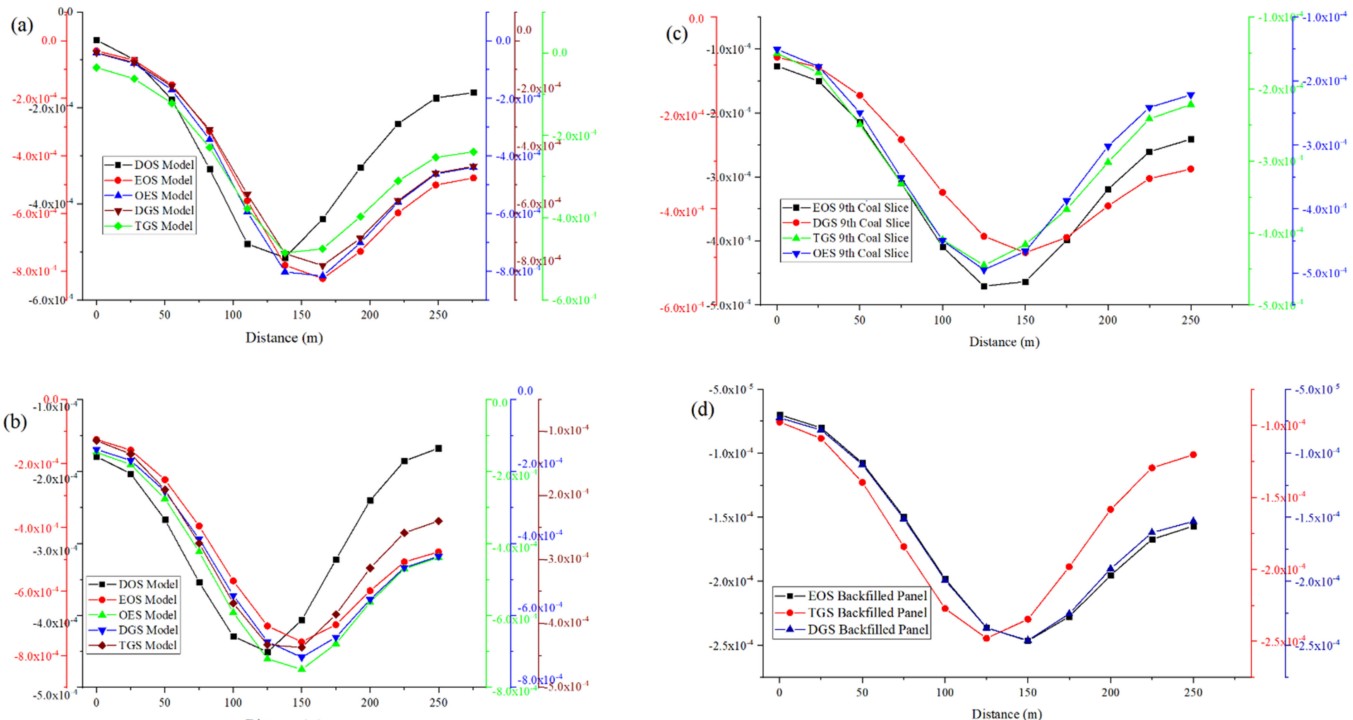

**Figure 12.** Vertical displacements. (**a**) Immediate roof displacement; (**b**) ground surface displacement; (**c**) 9th coal slice excavated and backfilled; (**d**) 12th coal slice excavated and backfilled.

## 5. Discussion

Amongst the five simulated models (DOS, OES, EOS, DGS, and TGS), the TGS shows the least level of induced ground displacement. The DOS model shows the most effective stress redistribution along the coal workface. It experienced the least shear deformation. However, the DOS is only practical with the application of web pillars and requires waiting for the setting of the backfill slurry.In addition, the shear stresses across the models show that the backfilled stopes are the zones with the highest shear deformations. This supports other research findings that the stress limits of the surrounding intact coal slice and rock are higher than the stress value of the backfill body.

Although the overburdens of the TGS and DGS models have relatively lower elastic strain energies, the TGS model requires excessive dismantling and transporting of the automated CMCB equipment, due to the long distance between coal slices to be mined. In the first excavation cycle, the DGS model (ground displacement 72 mm) shows that its two coal slices acting as temporary pillars do not significantly reduce vertical displacement compared with the EOS model (75 mm).

These drawbacks will mean that the DOS and TGS models are less desirable for site application, due to lower mineral recovery and operational time loss. Hence, the EOS or OES model is more suitable given its effective stress redistribution, non-requirement of web pillars and the fact that there is no excessive dismantling of CMCB equipment. The DGS method's ground displacement does not reasonably compensate for the time required to transfer mining equipment over a longer distance.

Therefore, the EOS and OES models show that a single intermediate coal slice is sufficient to act as a temporary pillar and retaining wall for the backfill slurry, without the interruption of the excavation operation and excessive dismantling and transfer of CMCB equipment. Since the EOS and OES models have 75 mm and 74 mm ground displacement, respectively, the OES is the most suitable mining sequence among the five models for the automated CMCB method.

## 6. Conclusions

(1) The proposed automated continuous mining continuous backfill (CMCB) method has mechanically integrated equipment such as the hydraulic support system, coupled with addcars using a telescopic cylinder, continuous miner, decouple machine, and a continuous haulage system.

(2) The model simulations show that induced ground displacements for a particular panel excavation will not continuously increase throughout the excavation operation, but peak when the backfill bodies become active in bearing the overlying load and the ground displacement will start to reduce and stabilize.

(3) The slice mining sequence-variation simulation shows that the odd-even sequence (OES) is the most efficient mining slice sequencing among the five slice variation methods. This is due to its ground displacement, non-requirement of web pillars, effective stress redistribution along the workface and less requirement for dismantling and transporting of mining equipment.

**Author Contributions:** Conceptualization, A.J.S.S.; Methodology, L.M. and S.A.A.; Software, S.A.A.; Validation, L.M. and A.J.S.S.; Formal Analysis, S.A.A.; Investigation, S.A.A.; Resources, L.M.; Data Curation, S.A.A.; Writing—Original Draft Preparation, S.A.A.; Writing—Review and Editing, L.M. and A.J.S.S.; Visualization, L.M.; Supervision, L.M. and A.J.S.S.; Project Administration, L.M. and A.J.S.S.; Funding Acquisition, L.M. All authors have read and agreed to the published version of the manuscript.

**Funding:** This paper was supported by the National Natural Science Foundation of China (51874280) and the Fundamental Research Funds for the Central Universities (2021ZDPY0211).

**Data Availability Statement:** All data supporting the findings in this study are available from the corresponding author on reasonable request.

**Conflicts of Interest:** The authors declare that they have no known competing financial interests or personal relationships that could have appeared to influence the work reported in this paper.

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
