# Peer review of "Ground Stress Analysis and Automation of Workface in Continuous Mining Continuous Backfill Operation"

_minerals, doi:10.3390/min12060754_

Round 1

Reviewer 1 Report

This paper examines the continuous mining and continuous backfill (CMCB) method and undertakes a CMCB numerical simulation study with 5 different sequences of coal excavation to determine the optimum sequence of resource excavation. The automated CMCB method has mechanically integrated equipment such as the hydraulic support system coupled with addcars using a telescopic cylinder, continuous miner, decouple machine, and a continuous haulage system.

The topic of the paper is very up-to-date and the numerical study carried out is relevant to be considered for publication in the journal Minerals. Moreover, the overall methodology is adequate regarding the main goals of the paper, and the main findings are well supported by the obtained results. I suggest the paper to be accepted for publication in its present form. The paper is clearly structured and very well written. The Introduction is very well written and organized, the description of the CMCB method is presented in a propped way, the results are very well presented and discussed, and the conclusions are well supported by the results achieved.

Author Response

Dear Reviewer,

Thank you for your time and the comment shared about our paper.

Reviewer 2 Report

This study proposes a modified mining method that enables automation at the face with minimal displacement while allowing higher flexibility in the excavation of resources at a lower capital cost. The paper needs improvement before it can be considered for publication. See my comments in attachment for improvement.

Reviewer 3 Report

Respected Authors

You considered a very important topic of the preservation of the daylight surface by controlling the stress-strain state of the massif during underground mining. For this purpose, you propose to apply the technology of sequential excavation of coal panels with subsequent backfilling in a coal mine.

I would like to note that the manuscript is very well written despite some minor shortcomings, that should be corrected to improve the manuscript and make it more Reader-friendly. Please find a list of my concerns and detailed comments below.

From my point of view, the current State of the Art in literature review is rather frugally presented. It is comprehensive indeed. The list of references consists of only 24 references. The Authors refer mainly to works of domestic Chinese scientists. Undoubtedly, Chinese scientists in recent decades have made a huge contribution to the study of the stress-strain state of the massif in the mining area. However, rich world experience has been accumulated in this field of science and the increase the geography of citation and use the research of scientists from other countries would be appreciated. I would not dare to impose you any particular reverences but I believe that some recent contributions might widen the perception of your study and raise its citing potential. Please check: https://doi.org/10.3390/su13116204 (Saudi Arabia), https://pmi.spmi.ru/index.php/pmi/article/view/15637/15645 (Russia), https://doi.org/10.1088/1755-1315/684/1/012007 (Poland), https://doi.org/10.1007/s10706-020-01586-x (Canada), and try a fast survey in Scopus search engine looking for more international references. 

I wonder if you could identify more clearly the problems raised in your study. It would be nice to summarize the main issues that other scientists have previously disclosed (or the main thing that is silent throughout the entire text of the manuscript. why?), maybe they did not directly address this problem (and why?), but it can be traced in works in related topics , and then bring to the tasks of the article itself - to close the blank spot.

I noticed that the Authors often express very correct thoughts, but in rather long sentences. As a result of overloading the sentence, the reader may lose the main idea. For example the sentences in lines 29-32; 34 - 37; 39-42 and so on. This list can be continued indefinitely throughout the text. From my point of view (as a non-native English speaker), such sentences might be divided into two or more, which will improve the perception of information by the Reader. Please consider that aspect.

In line 28, the authors make a statement, but do not support it with a link. In this case, lines 26-27 are confirmed by reference [1]. I think that you need to add a link.

In lines 38 - 42, the authors reveal the problems that arise when extracting a mineral by an underground method. At the same time, only two sources are cited. It should be noted that for each of the problems named by the authors, more than a hundred studies have been carried out by scientists from Serbia, Slovakia, Russia, Poland, Spain and others. From my point of view, it is necessary to turn to these works, give a brief analysis of them, and identify unresolved issues in these studies. Or say what prevents the results of these studies from being applied to the Changxing coal mine in China.

Same problem appears in lines 50-53. There is no analysis of world experience. What has not been done so far or what prevents it from being applied to the conditions of the mine under study.

From my point of view, it is necessary to briefly summarize the analysis of previous works and identify the issue facing the authors for resolution in this study. With a fairly clear statement of the purpose of the study (line 65-68), the authors did not identify the objectives of this work. Eliminating these shortcomings will allow the reader to more accurately understand the direction of this study.

From my point of view, again. in the section "Materials and Methods" it is necessary to give brief information (possibly in a table) about the geological structure of the deposit; depth of development (mining operations); chamber (panel) length; its height; breaking method and other information that will allow other researchers to understand (repeat) the experiment. The authors made a reference to the source [17] from which some of the information was taken. But this is not enough, since the information on the experiment should be in front of our eyes.

It is necessary to indicate who made the drawings 1-7. If by the authors, then indicate in parentheses (performed by the authors) and with what program. If borrowed, then you need to make links and have permission from the copyright holder.

In section 2.2. the authors argue about the minimization of mine ventilation control as a result of the transition to unmanned mining technology. At the same time, coal mines are very dangerous in terms of explosions of accumulated methane. It is necessary to indicate what measures are envisaged to control methane emissions and reduce its concentration in the mine.

In Chapter 2, the authors describe the "Automated Continuous Mining Continuous Backfill Method" in sufficient detail. From my point of view, again, such a detailed description does not make sense, since this is not a research question. As mentioned earlier in note it is necessary to provide brief information, possibly in the form of a table and a brief explanation with figures, which will greatly improve perception. But I may be mistaken.

What is figure 4 for? As I understand it, this is a standard support design and the authors do not modernize it. From my point of view, there is no need to provide publicly known information, as well as detailed information in section 3.3. Table 1 with a brief description is sufficient. If this is still a modified version of the serial version, then it is necessary to indicate what changes have been made. If changes are made by other researchers, then provide a link.

What is figure 5 for? The same comment as in previous remark.

As in the previous remarks about figures 6 and 7 and the descriptive part to them. Well-known information overloads the text and distracts from the main idea.

From the title of the article, it becomes clear that the authors propose a mining technology for research with subsequent backfilling. Only from the text it is not clear: with what material the authors propose to lay the waste space; how the filling material is prepared and how this filling material will be fed into the worked-out space.

The authors did not indicate clearly the boundary values ​​for numerical modelling in FLAC, which makes it impossible to repeat the study.

Dear Authors

I have to apologise for a very direct way of reviewing your paper and addressing issues that I find to be shortcomings. As a non-native English speaker I'm slightly worried if my comments do not sound offensive. It has never been my intention as I appreciate very much your work.

Sincerely

Round 2

Reviewer 2 Report

Re-submit the paper with highlighted changes and also prepare a response to reviewers, ensuring that all comments/queries are addressed.

Author Response

Dear Reviewer,

Thank you for the corrections and insight shared. All queries have been answered, please see the attachment. The modified texts are highlighted in red in the resubmitted manuscript.

Kind regards

Reviewer 3 Report

Respected Authors   I appreciate your study very much but I'm very much confused with your attitude to Reviewers' comments. Usually, reviewing for MDPI journals, I expect "step by step" responses to my comments (even arguing when addressing my concerns - no problem). So far, you did not really respond to my comments, nor your responses were reflected in the revised version of the manuscript. Even editorial issues are still very messy in the body text and especially in the reference list which is not formatted to the MDPI template.   Dear Authors I believe (without any doubts) that your study should be considered for publication in Minerals, and may be easily corrected (developed) with a little effort from the Authors' side.   That is why I maintain my previous assessment (major revision).   Sincerely yours 

Author Response

Dear Reviewer,

I sincerely apologise if any of my earlier comments seem rude, especially on arguing with you. I have provided a step by step response to your comments, please see the attachment. More importantly, the introduction has been improved for clarity on the study's objective.

Kind regards

Round 3

Reviewer 2 Report

The authors may remove any overlapping lines and text from the figures. The number of figures may be reduced by combining a few. Proofread to improve the reading. Improve and expand discussions. Add more relevant references if possible.

Author Response

Dear Reviewer,

Thank you for the corrections and comments shared on improving the manuscript. Please see the attachment.

Reviewer 3 Report

Respected Authors

I accept your answers, however I still support my position. The diversity of opinion is a value in scientific world.

I opt for a minor revision, concerning mainly format of references that is not in accordance with MDPI template.

Sincerely yours 

Author Response

(The authors gave the same response as above.)
